# Humidity Sensing Properties of (In+Nb) Doped HfO_2_ Ceramics

**DOI:** 10.3390/nano13050951

**Published:** 2023-03-06

**Authors:** Jiahao Yao, Jingsong Wang, Wenjun Cao, Li Li, Mingxiang Luo, Chunchang Wang

**Affiliations:** Laboratory of Dielectric Functional Materials, School of Materials Science & Engineering, Anhui University, Hefei 230601, China

**Keywords:** hydrothermal method, impedance humidity sensor, co-doping, oxygen vacancies

## Abstract

(In+Nb) co-doped HfO_2_ ceramics, Hf_1-*x*_(In_0.5_Nb_0.5_)*_x_*O_2_ (*x* = 0, 0.005, 0.05, and 0.1), were prepared via a solid-state reaction method. Dielectric measurements reveal that the environmental moisture has an obvious influence on the dielectric properties of the samples. The best humidity response was found in a sample with the doping level of *x* = 0.005. This sample was therefore selected as a model sample to further investigate its humidity properties. In doing so, nanosized particles of Hf_0.995_(In_0.5_Nb_0.5_)_0.005_O_2_ were fabricated via a hydrothermal method and the humidity sensing properties of this material were studied in the relative humidity range of 11–94% based on impedance sensor. Our results show that the material exhibits a large impedance change of nearly four orders of magnitude over the tested humidity range. It was argued that the humidity-sensing properties were related to the defects created by doping, which improves the adsorption capacity for water molecules.

## 1. Introduction

Humidity sensors are known to drive extensive research due to their widespread use in medical diagnostics, the automotive industry, agricultural activities, and some special circumstances such as wet baby diaper, skin humidity, and spatial positioning monitoring [1,2]. The types of humidity sensors include resistance sensors, capacitance sensors, impedance sensors, pressure sensors, and light/intensity sensors [3,4]. Among these, the impedance humidity sensor has been widely used in recent years because of its convenient and sensitive characteristics. It has been reported that a large number of ferroelectric materials are candidates for high sensitivity humidity sensitive materials, such as Na_0.5_Bi_0.5_TiO_3_, K_0.5_Bi_0.5_TiO_3_, and K_0.5_Na_0.5_NbO_3_ [5,6,7]. The structure of these ferroelectric materials is relatively complex and the preparation processes are difficult. Therefore, it is necessary to find a simple method to synthesize the ferroelectric humidity sensitive materials with relatively simple structure.

As a promising high-k material, HfO_2_ has become a research hotspot in the microelectronics industry in recent years [8]. It had been reported that ferroelectricity can be induced in hafnium oxide films by slightly doping with specific elements such as Gd, Al, Y, and Si [9,10,11,12,13]. This behavior provides the material tremendous promise for the applications of ferroelectric dynamic random-access memory, ferroelectric field effect transistors, ferroelectric tunnel junctions, and negative capacitance devices [14,15,16]. From the viewpoint of application, a fundamental understanding and then manipulating of the dielectric properties of HfO_2_ are of vital importance. In addition, HfO_2_ is one of the few dioxides with both ferroelectric properties and a large band gap of 5.7 eV [17]. Currently, donor–acceptor co-doping has been proved to be an effective and facile strategy to modify the dielectric properties of oxides [18] and has been widely used in the TiO_2_ system [19,20,21,22,23]. Additionally, this strategy has also been extended to other oxides [24,25,26,27,28,29]. Nowadays, two main features can be extracted in co-doped oxides: colossal dielectric properties as reported in most co-doped oxides, or humidity sensitive effects reported in a few co-doped oxides, such as (Na+Nb) co-doped TiO_2_ [30], (In+Nb) co-doped SnO_2_ [28]. So far, there are no experimental studies on the dielectric properties of donor–acceptor co-doped HfO_2_ ceramics. This motivates us to investigate the dielectric properties of (In+Nb) co-doped HfO_2_. In addition, the morphology and structure over a humidity sensing material also play an important role in its humidity performance. Compared with micro-sized bulk materials, materials with nano-sized structure can provide more active sites for adsorbing water molecules, thereby greatly improving humidity sensing properties. To our surprise, (In+Nb) co-doped HfO_2_ ceramics show excellent humidity properties instead of colossal dielectric behavior.

Herein, the (In+Nb) co-doped Hf_1-*x*_(In_0.5_Nb_0.5_)*_x_*O_2_ (*x* = 0, 0.005, 0.05, and 0.1) ceramics were successfully prepared via a solid-state reaction method. The sample of Hf_1-*x*_(In_0.5_Nb_0.5_)*_x_*O_2_ (*x* = 0.005) shows the largest response and a large impedance change of nearly four orders of magnitude. The humidity sensing properties can be attributed to the dual doping-induced defects, which act as active sites for water absorption. We, herein, firstly report the humidity properties and the dielectric properties of (In+Nb) co-doped HfO_2_ ceramics will be reported in an ensuing paper elsewhere.

## 2. Materials and Methods

In this work, both solid phase and wet chemical methods were used to fabricate the samples. First, Hf_1-*x*_(In_0.5_Nb_0.5_)*_x_*O_2_ (*x* = 0, 0.005, 0.05, and 0.1, abbreviated as HINO-0, 0.005, 0.05, and 0.1, respectively) were prepared via a solid-state reaction method in order to find out which doping level leads to the best humidity sensing properties. HfO_2_ (≥99.99%), Nb_2_O_5_ (≥99.99%), and In_2_O_3_ (≥99.99%) were used as starting materials. These powders were weighed according to stoichiometric compositions and were thoroughly mixed via ball milling for 12 h. The mixtures were calcined at 1473 K for 10 h. Then, the mixtures were ball milled again. The resulting powders were pressed into pellets with an average thickness of 0.80 mm using polyvinyl alcohol (PVA) as a binder. Finally, these pellets were sintered at 1673 K for 6 h. To test the humidity properties, copper wires were glued on both sides of the sintered pellets as electrodes. The results shown in Figure 5 reveal that the sample with *x* = 0.005 shows the best humidity properties. Since the humidity properties strongly depend on sample surface, materials with large specific surface area (SSA) are more conducive to the humidity properties. The sample HINO-0.005 was, therefore, selected to be further fabricated using a hydrothermal method in order to increase its SSA values. The flow diagram is shown in Figure 1. Firstly, 4.0000 g HfCl_4_ (99.9%), 0.0094 g InN_3_O_9_ (99.99%), and 0.0085 g NbCl_5_ (99%) were dissolved into a beaker containing 250 mL deionized water to form solution A. Secondly, 30 g NaOH was dissolved into another beaker containing 250 mL deionized water to form solution B. Next, 250 mL of solution A was added into solution B under magnetic stirring for 20 min. Thirdly, the mixed solution was transferred to a stainless-steel autoclave lined with polytetrafluoroethylene and heated at 393 K for 24 h. Finally, the obtained sample was repeatedly washed and centrifuged several times and dried in an oven at 323 K overnight.

The humidity sensor was fabricated via the aerosol deposition method using a spray gun (Sao Tome V130, Sao Tome Building Materials Co., Qingdao, China) with a diameter of 0.2 mm. Firstly, the HINO-0.005 powder was mixed with 10 mL absolute ethanol solution and sonicated for several minutes to form uniform paste. Secondly, 10 mL of slurry was uniformly sprayed on Al_2_O_3_ substrate covered with Au interdigital electrodes. Finally, the two copper wires were glued on the Au interdigital electrode with silver paste and dried at 373 K for 16 min.

The phase structure and microstructure of the samples were tested by X-ray diffraction (XRD, Rigaku Smartlab Beijing Co., Beijing, China) and scanning electron microscope (SEM, Model S-4800, Hitachi Co., Tokyo, Japan) equipped with energy-dispersive spectroscopy (EDS), respectively. The humidity sensing properties were measured with a Hioki 3532-50 LCR HiTester (Tokyo, Japan) precision impedance analyzer with different frequencies. The amplitude of the AC signal is 100 mV. Different humidity environments were provided by supersaturated LiCl, MgCl_2_, Mg(NO_3_)_2_, NaCl, KCl, and KNO_3_ salt solutions, which yield relative humidity (RH) levels of 11, 33, 54, 75, 85, and 94%, respectively, at room temperature, as shown in Figure 2. A RT6000 ferroelectric tester (Radiant technologies Inc., Albuquerque, NM, USA) was used to study the polarization-electric field (P-E) loop at 1 kHz and room temperature.

## 3. Results and Discussion

### 3.1. Ceramic Samples

Figure 3 is the XRD patterns of the ceramic samples. It is seen that all samples exhibit pure phase, and the diffraction peaks can be indexed according to PDF#74-1506. Figure 4a–d display the thermally etched surface SEM images of the ceramic samples HINO-0, 0.005, 0.05, and 0.1, respectively. The samples show well identified grains with a small number of pores among these grains. The grain size strongly depends on the doping levels [31]. The statistical grain size distributions of the samples are displayed in Figure 4e–h. As the (In_0.5_Nb_0.5_) concentration gradually increases from *x* = 0 to 0.1, first, the average grain size of the ceramics gradually decreases from 0.32 to 0.2 μm, then it gradually increases from 0.2 to 0.5 μm, indicating that the (In_0.5_Nb_0.5_) doping in the HfO_2_ matrix can effectively align grain growth. The average grain sizes were found to be 0.32, 0.20, 0.38, and 0.50 μm for HINO-0, 0.005, 0.05, and 0.1, respectively.

The humidity properties of the ceramic samples were displayed in Figure 5. Two features can be extracted from the figure: (1) all samples are almost humidity insensitive in the low humidity range, but strongly humidity sensitive in the high humidity range, regarding this phenomenon, this is due to the relatively small specific surface area of the sample, which has a relatively small contact area with water molecules, and the associated adsorption processes are not evident in low and medium humidity environments. Only at ambient levels, due to the porous structure of the sample, does capillary condensation of water molecules occur in the pores, which makes the impedance of the material start to decrease significantly; (2) the HINO-0.005 sample shows the maximum impedance variation with impedance decreasing from 7.151 × 10^7^ Ω @11%RH to 2.955 × 10^5^ Ω @94%RH found from the curve recorded with 100 Hz. Therefore, the sample was selected as a model sample. The hydrothermal method was used to prepare the sample in order to reduce its particle size to nanoscale. Before doing so, ferroelectric properties of this sample were examined as ferroelectric properties were frequently reported in slightly doped HfO_2_ film [32,33,34,35,36,37]. The P-E loop and corresponding current-voltage (I-V) curves of the HINO-0.005 are shown in Figure 6a. The P-E loop is olive-shaped as seen in Figure 6a, which is not the case of the electric hysteresis loop in thin-film materials [38,39,40,41,42]. The large amount of polarized charges on the surface of bulk ceramics may be the main reason for this phenomenon.
Figure 5(**a**–**d**) Impedance as function of RH level of the ceramic samples.
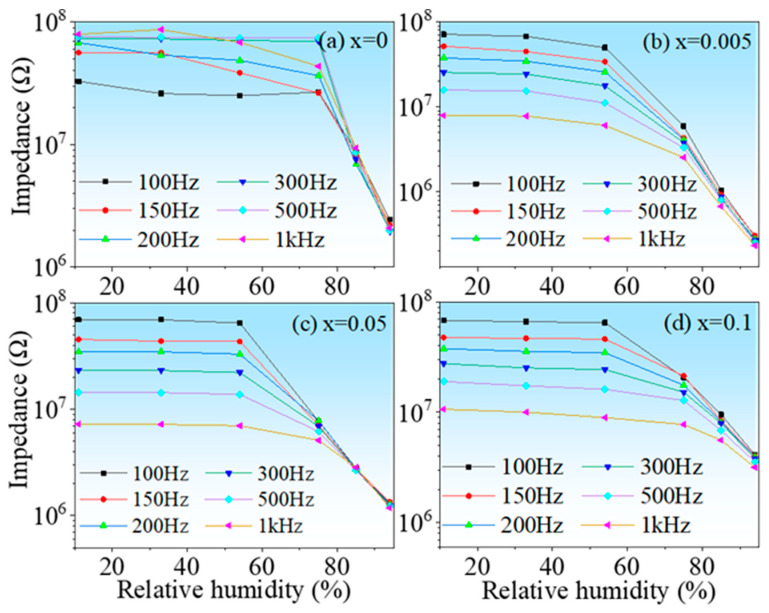



### 3.2. Nanomaterial of HINO-0.005

An XRD pattern of the HINO-0.005 sample prepared via the hydrothermal method is shown in Figure 6b. Compared with those prepared with the solid-state reaction method, the reflection peaks are much broader, confirming the nano-sized structure of the present sample. The inset on the left of Figure 6b is a photograph of the sensor, and the inset on the right of Figure 6b is the SEM image of the HINO-0.005 nano-powder. Figure 6c shows the EDS spectrum of HINO-0.005. From the figure, we can see the presence of Hf, O, Nb, and In elements and the atomic percentages of the elements are shown in Table 1.

To demonstrate that In and Nb elements are doped into the HfO_2_ lattice, we performed elemental mapping of In, Nb, Hf, and O as shown in Figure 7a–d, respectively. It can be seen from Figure 7 that In and Nb as doping elements show good uniformity in the maps. Therefore, we believe that In and Nb elements are successfully doped into the lattice of HfO_2_.

Figure 8a shows that the impedance of the sensor decreases significantly with increasing RH level without the low-RH level platform as observed in the solid-method-prepared samples [please see Figure 5]. Interestingly, the impedance variation is one order of magnitude larger than that of the solid-method-prepared samples. With the increase in frequency, the range of impedance variation of the sensor appears to decrease significantly. In particular, the greatest change in impedance occurs at a frequency of 100 Hz. This is because the polarization of the water molecules cannot keep up with the electric field changes at high frequencies. Additionally, the impedance recorded at 100 Hz almost linearly varies with the humidity. Therefore, the frequency 100 Hz was chosen as the working frequency for the subsequent work.

The humidity sensing response is calculated using the following relation [43,44]:(1)$$S=Z_{d}/Z_{h}$$
where the impedance values measured at 11%RH and a specific RH level are $Z_{d}$ and $Z_{h}$, respectively. The calculated S values of the sensor are listed in Table 2. The largest S value of 3612.880 was achieved. Although the humidity response S value does not meet the requirements for a high-performance humidity sensor, this result underscores that the slightly (In+Nb) co-doped HfO_2_ exhibits obvious humidity sensitivity.

The hysteretic behavior of the sensor is evaluated in terms of impedance changes during adsorption and desorption processes. In all humidity environments, the sensor was kept for 5 min and then the changes in impedance were recorded; the results are shown in Figure 8b. The hysteresis value is calculated based on the relation [45]:(2)$$\left[\frac{\log(Z_{\mathrm{ads}})-\log(Z_{\mathrm{des}})}{\log\left(Z_{\mathrm{ads}}\right)}\right]\times 100\%$$
where $Z_{\mathrm{des}}$ and $Z_{\mathrm{ads}}$ are the impedance values of the desorption and adsorption processes. Table 3 shows the Z_des_ and Z_ads_ and hysteresis values of the HINO-0.005-based sensor in the range of 11–94%RH. The sensor shows small hysteresis value in various humidity environments.

The repetitive behavior of the sensor is shown in Figure 8c. Sensor repeatability was achieved by alternating cycles of 5 times between RH = 11% and RH = 94% environments. In each environment, the sensor is exposed for 300 s to adapt to the change in ambient humidity. The sensor switching time between different humidity environments is negligible compared to the 300 s of exposure in each environment, because the switching is done in a very short time (roughly 1 s). According to the test results, we found that the impedance of the sensor remains basically stable, so it can be concluded that the sensor has good repeatability. The recovery/response characteristics of the sensor were measured by placing the sensor in alternating 11 and 94% humidity environments with a duration time of 5 min in each humidity environment. The response–recovery curve is shown in Figure 8d. The response/recovery time is the time required for the impedance change to reach 90% of the total impedance change during adsorption/desorption process. The response/recovery time of the sensor is found to be 20/50 s, indicating that the sensor has good recovery–response behavior. Compared with the adsorption process, the desorption time is longer, which is due to the higher moisture-sensitive response of adsorbed water molecules and the surface of the sensor material as well as the higher binding energy. In addition, the adsorption process of water molecules is an exothermic process, while the desorption process is a heat absorption process requiring external energy to make water molecules escape from the sensor surface. In addition to the above factors, the hysteresis factor is also related to the amount of sensing material and the diffusion rate of water molecules [46,47].

It has been reported that the environmental conditions, such as experimental environment (isobaric measurement), actual environment (measurement in open atmosphere), and corrosive environment (measurement in different concentrations of corrosive ionic salt solutions), have strong influence on the sensitivity of the sensor. Among them, in the experimental and actual environments, the environmental conditions have less influence on the sensor sensitivity with time. However, in corrosive environments, a greater degree of reduction in sensor sensitivity occurs over time [48]. Based on this consideration, the long-term stability of the sensor was tested. Figure 9a and Figure 9b show, respectively, the impedance and sensitivity of the sensor as a function of ambient humidity measured every 5 days. It is found that both the impedance and sensitivity decrease significantly with time. This is mainly due to the reaction between the HINO-0.005 nanoparticles and the corrosive ions volatilized in the corrosive environment, and the corrosion layer on the surface of the nanoparticles, which caused irreversible damage to the sensor. From Figure 9c,d, it can be seen that, compared with 85%RH, the sensitivity of the sensor decreases more significantly with time at 94%RH. This can be attributed to the more intense reaction of the corrosive ions with the nanoparticles on the sensor surface at higher ambient humidity levels.

### 3.3. Humidity Sensing Mechanism

Since the impedance spectrum can provide more information about the sensing mechanism of humidity sensor [6,49], impedance analysis was performed on the present sensor. The complex impedance diagrams for three different humidity ranges are shown in Figure 10. From the complex impedance plots, one notes that the complex impedance plot of the sensor is a semicircle with a large radius at low humidity levels, as shown in Figure 10a. When the humidity level increases to 75%, the radius of the semicircle undergoes a significant reduction, as shown in Figure 10b. At the highest humidity levels of 85 and 94%RH, the radius of the semicircle continues to reduce but followed by a straight line at the end of the semicircle, as shown in Figure 10c. The impedance results can be explained as follows:

The purpose of (In, Nb) co-doping is to construct electronically pinned defect dipoles in the metal oxide. Nb^5+^ as a donor can provide a lot of delocalized electrons, as described by the defect equation:(3)$${\mathrm{Nb}}_{2}O_{5}\overset{2{\mathrm{HfO}}_{2}}{\to }2{\mathrm{Nb}}_{\mathrm{Hf}}^{•}+4O_{O}+1/2O_{2}+2e^{\prime }$$

Likewise, In^3+^ as an acceptor will generate a lot of oxygen vacancies, providing a local anoxic environment to inhibit the delocalized electrons, resulting in a charge-balanced defect-dipole complex/cluster.
(4)$${\mathrm{In}}_{2}O_{3}\;\overset{2{\mathrm{HfO}}_{2}}{\to }2{\mathrm{In}}_{\mathrm{Hf}}^{\prime }+3O_{O}+V_{O}^{••}$$

Due to the high electrical properties of Hf atoms, discontinuous hydrogen bonds are formed after the water molecules are trapped by the Hf edges. At low humidity levels (RH = 11%, 33%, and 54%), the content of water molecules is very low and water molecules are adsorbed in the form of chemisorption and cannot move freely. A continuous water layer is not yet formed on the sensor surface, and the conduction of free protons from one position to another on the surface is difficult, easily leading to the generation of high impedances, as shown in Figure 11. This can also be confirmed in the complex impedance diagram. At 11–54%RH, the complex impedance diagram of the sensor exhibits a large semicircle, as seen in Figure 10a. The appearance of the semicircle is a result of the proton current conduction, which can be equated to the parallel connection of a resistor and a constant phase element CPE [50]. At low humidity, the proton current conduction is limited, mainly because the movement of the proton current at the surface requires crossing an energy barrier. Therefore, the humidity sensor exhibits a large impedance in this humidity range. Moreover, at low humidity, where CPE is mainly the dominant factor, the impedance of the sensor is more susceptible to frequency. This is also confirmed as shown in Figure 8a.

Depending on the defect response, oxygen vacancies can also cause humidity sensing in the form:(5)$$H_{2}O+V_{O}^{••}+O_{O}\to 2{\mathrm{OH}}_{O}^{•}$$

When RH = 75%, from the complex impedance diagram, a significant decrease in impedance occurs and a smaller semicircle is formed, as shown in Figure 10b. This is due to the oxygen vacancy as the active center of water absorption, which generates a large amount of electrostatic OH^−^ group. The electrostatic field generated by the ${\mathrm{Nb}}_{\mathrm{Hf}}^{•}$ defect can also dissociate water molecules to produce OH^−^ groups. They can also act as an effective chemisorption centers and play a role in reducing the sensor impedance.

Due to the electrostatic effect of OH^−^ groups, more and more water molecules are physically absorbed to form a continuous water layer [51]. The formation of a continuous water layer can facilitate the combination of water molecules and OH^−^ groups to form H_3_O^+^ and, at the same time, H_3_O^+^ can also generate H^+^ and H_2_O.
(6)$$H_{3}O^{+}\to H^{+}+H_{2}O$$

This process promotes the transfer and hydration of H_3_O^+^, which greatly reduces the impedance of the sensor and improves the sensitivity of the sensor.

With the further increase in humidity level, the relative humidity reaches above 85%, and the formation of multiple continuous layers of water due to physical adsorption makes the physical adsorption close to saturation. At this point, the decrease in sensor impedance is caused by the combined jump of H_3_O^+^ and protons. When H_3_O^+^ releases a proton to a neighboring water molecule, the latter accepts the proton and releases another proton at the same time, and so on repeatedly; charge transfer occurs, which is called Grotthuss chain reaction. It is considered to represent the conduction mechanism of liquid water. According to Grotthuss [52] and Casalbore-Miceli [47]:(7)$$H_{2}O+H_{3}O^{+}\to H_{3}O^{+}+H_{2}O$$

The impedance plot shows a straight line at the end of the semicircle. This is due to the saturation of the physical adsorption and the Warburg impedance caused by the electroactive material on the electrode [53]. Moreover, the frequency dependence of the impedance is reduced due to the presence of the Warburg impedance, as shown in Figure 10c. Therefore, at high RH levels, the Grotthuss mechanism plays a dominant role, accounting for the humidity sensing properties of the (In+Nb) co-doped HfO_2_ ceramics.

## 4. Conclusions

In conclusion, (In+Nb) co-doped, Hf_1-*x*_(In_0.5_Nb_0.5_)*_x_*O_2_ (*x* = 0, 0.005, 0.05, and 0.1) ceramics were prepared via the solid-state reaction method. The humidity-sensing properties of the samples were investigated over the humidity range of 11–94%RH. The sample with *x* = 0.005 was found to show the largest response, and was used as a model sample to further boost its sensing properties via the wet chemical method to reduce the particle size and enlarge the specific surface area of the sample. A large impedance change of nearly four orders of magnitude was achieved in the sensor based on nanosized HINO-0.005. The humidity sensing properties were attributed to the dual doping-induced defects that act as active sites for water absorption. The Grotthuss mechanism can be used to explain the humidity sensing properties of the (In+Nb) co-doped HfO_2_ ceramics.

## Figures and Tables

**Figure 1 nanomaterials-13-00951-f001:**
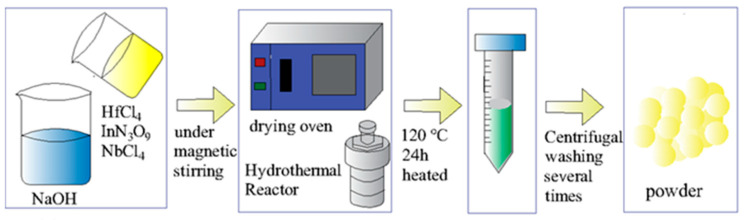
Hydrothermal route for the synthesis of HINO-0.005 powder.

**Figure 2 nanomaterials-13-00951-f002:**
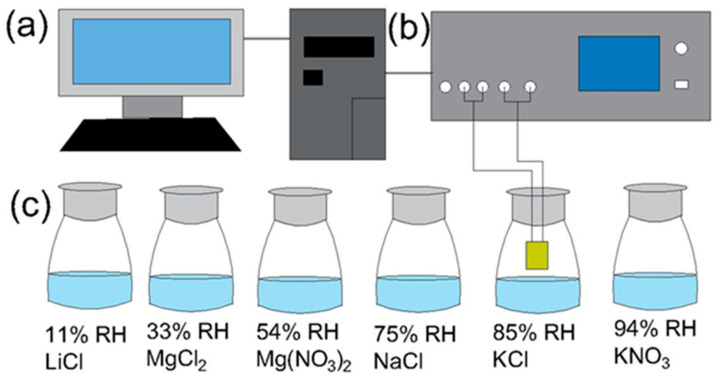
Schematic of the humidity sensing experimental setup: (**a**) PC; (**b**) Hioki 3532-50 LCR; and (**c**) different humidity environments.

**Figure 3 nanomaterials-13-00951-f003:**
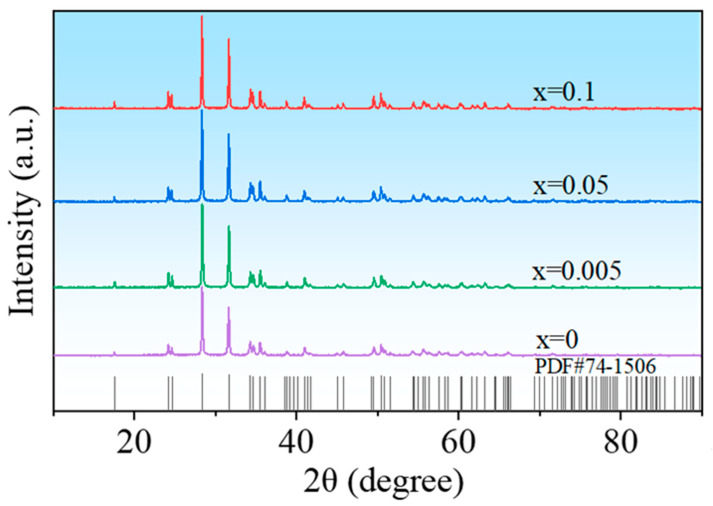
XRD patterns of the ceramic samples prepared via the solid-state reaction method.

**Figure 4 nanomaterials-13-00951-f004:**
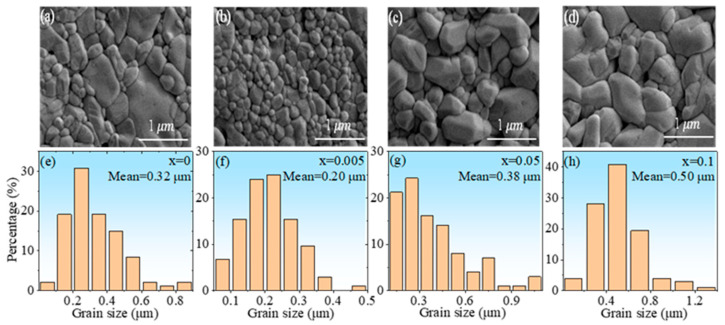
(**a**–**d**) SEM images of ceramic samples and (**e**–**h**) the corresponding particle size distributions.

**Figure 6 nanomaterials-13-00951-f006:**
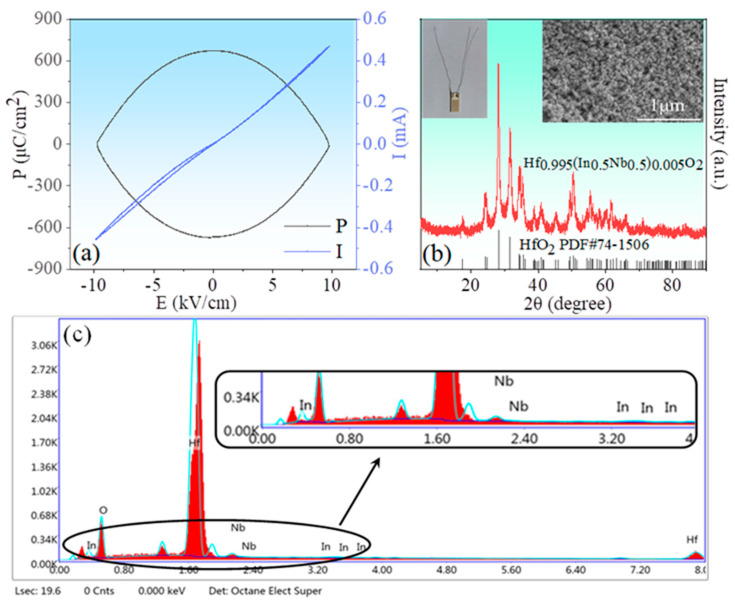
(**a**) P-E loop and I-V curves of HINO-0.005; (**b**) XRD patterns (main panel), SEM image (right inset), and picture of the sensor (left inset) of HINO-0.005; (**c**) EDS spectrum for HINO-0.005.

**Figure 7 nanomaterials-13-00951-f007:**
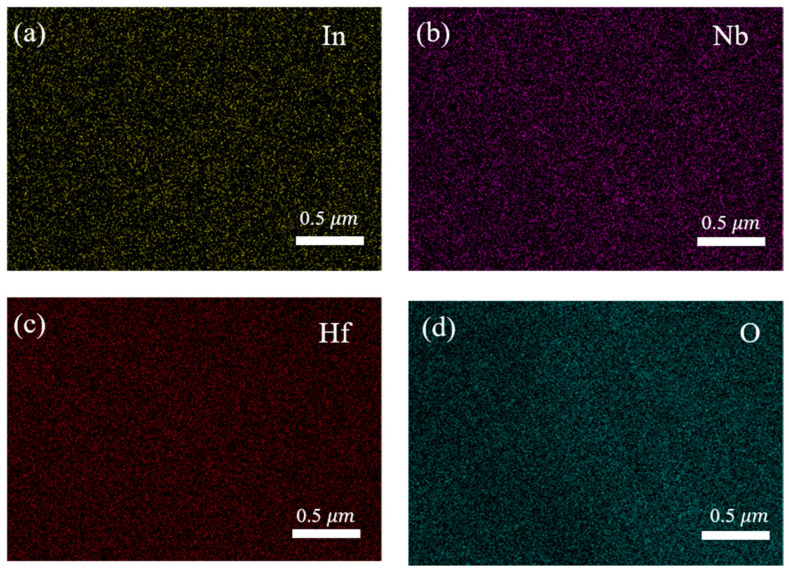
Elemental mappings of (**a**) In, (**b**) Nb, (**c**) Hf, and (**d**) O.

**Figure 8 nanomaterials-13-00951-f008:**
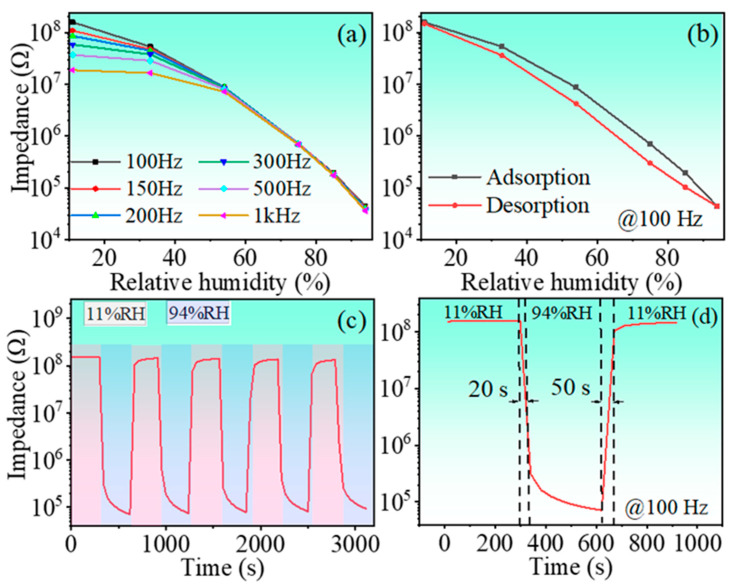
(**a**) Impedance as function of RH level, (**b**) humidity hysteresis curve, (**c**) repeatability curve, and (**d**) response/recovery curve of the HINO-0.005.

**Figure 9 nanomaterials-13-00951-f009:**
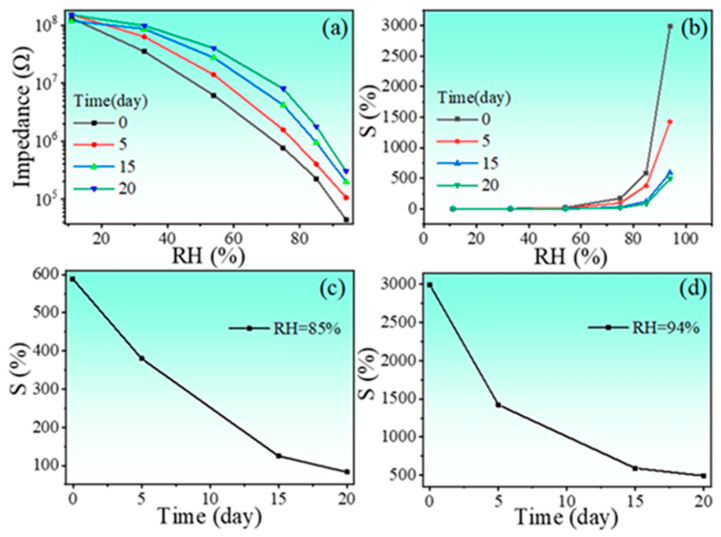
Variations in sensor impedance (**a**) and sensitivity (**b**) relative to humidity for different time intervals; Variations in sensor sensitivity with time at 85 (**c**) and 94%RH (**d**).

**Figure 10 nanomaterials-13-00951-f010:**
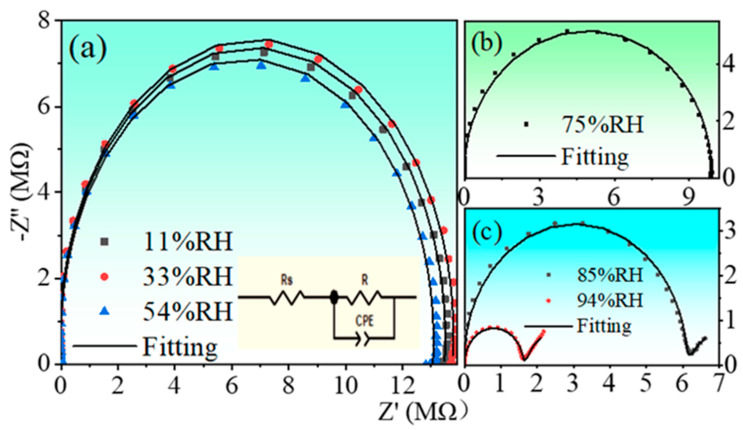
Complex impedance diagrams of the sensor based on nanosized HINO-0.005: (**a**) low RH levels (11, 33, and 54%); (**b**) medium RH level (75%); (**c**) high levels (85 and 94%). The inset shows the equivalent circuit used for data fittings.

**Figure 11 nanomaterials-13-00951-f011:**
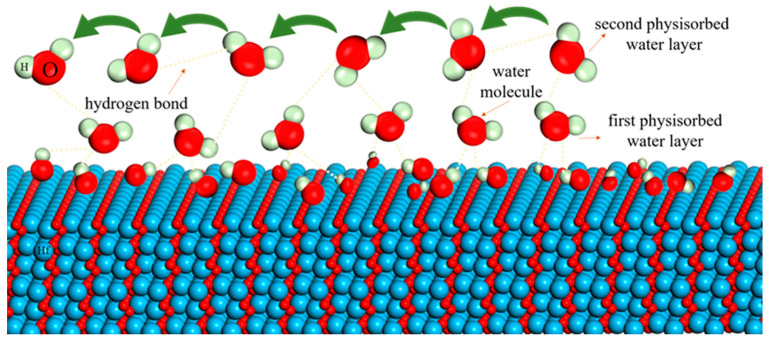
The sensing mechanism diagram of the sensor.

**Table 1 nanomaterials-13-00951-t001:** Local elemental compositions of the HINO-0.005 deduced from EDS.

Element	Weight %	Atomic %
O	20.60	73.93
Nb	1.63	1.01
In	0.21	0.11
Hf	77.56	24.96

**Table 2 nanomaterials-13-00951-t002:** Impedance and *S* values of the sensor based on nanosized HINO-0.005.

RH (%)	Impedance (MΩ)	*S* (Z_d_/Z_h_)
11	158.3	1
33	53.262	2.972
54	8.654	18.285
75	0.705	224.711
85	0.192	824.283
94	0.044	3612.880

**Table 3 nanomaterials-13-00951-t003:** Humidity hysteresis values of the sensor based on nanosized HINO-0.005.

RH (%)	Z_ads_ (Ω)	Z_des_ (Ω)	Hysteresis Values (%)
11	1.583 × 10^8^	1.428 × 10^8^	0.351
33	5.326 × 10^7^	3.599 × 10^7^	2.25
54	8.654 × 10^6^	4.217 × 10^6^	4.71
75	7.046 × 10^5^	2.992 × 10^5^	6.79
85	1.921 × 10^5^	1.023 × 10^5^	5.46
94	4.382 × 10^4^	4.382 × 10^4^	----

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
