# Peer review of "Humidity Sensing Properties of (In+Nb) Doped HfO2 Ceramics"

_nanomaterials, 2023, doi:10.3390/nano13050951_

Round 1

Reviewer 1 Report

There are some elements that the authors should clarify in the manuscript.

- First of all, it should be justified more applied the choice of minimal substitutions (up to 0.1).

- Impedance measurements must be detailed with the specification of at least the frequency of the alternative electrical signal used.

- It should be mentioned in the manuscript that the salt solutions used to establish different humidity levels vaporize not only the water from the solution but also salts that change the conductivity of the sensors differently compared to pure vapors (uncontaminated by salts). Especially semiconducting materials, in the present case, can induced conductivities much different from those exposed to pure water vapors etc. In addition, the salts once infiltrated into material pores disturb the subsequent measurements etc. For example, reduced sensitivity to low humidity levels (as can be seen from Fig. 5. those initial almost horizontal characteristics) can have such an explanation.  In this sense, some reproducibility measurements or at least an analysis of these aspects (the influence of corrosive environments etc.) would be useful (See https://doi.org/10.1007/s10854-018-9987-y)

- Regarding the quality of the presentation, the text must (re)written more carefully, the conclusions need to be a bit more elaborated etc.

Reviewer 2 Report

The article is clearly and concise written and is well understandable. It reports on an In,Nb-codoped HfO2 ceramics based impedance sensor of humidity. Some points need to be clarified before it can be published.

1) Materials characterization is insufficient in the article. The main goal was to obtained In,Nb-codoped HfO2, but no proofs of the successive doping can be found in the presented results. Were In and Nb incorparated into the lattice of HfO2, or maybe the additives were segregated as separate In2O3 and Nb2O5 species on the surface of HfO2? Due to low percentage of the additives, XRD is helpless to resolve this issue, since XRD has low sensitivity to small percentages of impurity phases. However, some information can be extracted from the refined unit cell parameters from XRD data. Elemental distribution map can also be helpful to verify that In and Nb are homogeneously distibuted and not segregated.

2) The impedance measurements should be described in more details. How much time does a single measurement take, so that the dynamic responses in Fig.7c,d  could be obtained? What about the time needed to change the bottles with different salts solutions in which the sensor is placed when switching between the different humidities? In Fig. 7 we see that time has a resolution of seconds.

3) Sensing mechanism needs clarification and detalization. What was the role of oxygen vacancies in adsorption of humidity (eq. 3)? Water can be adsorbed on cation-anion pairs (2 H2O + Hf4+ 2 O2- =  Hf(OH)4) at an oxide surface without any vacancies.

If In doping produces oxygen vacancies, no new vacancies are expected in In,Nb-codoped HfO2, since these dopants are compensating each other, as the authors also stated at the beginning. Comments of this.

The details of Grotthuss mechanism should be added and discussed, since the eq. 4 is meaningless: it has the same two compounds on the left and right sides.

What is the origin of energy barriers for protons to cross at the surface in presence of humidity?

What is the origin of hysteresis when increasing and decreasing humidity?

Also, the response dependences on humidity in Fig. 5 look like there is a critical humidity level below which there is no response and after which the response depends on humidity. It resembles an isotherm of adsorption (e.g. Langmuir). Maybe the reason was the capillary condensation of water in the materials pores which occured at a sufficiently high humidity?

Round 2

Reviewer 1 Report

Ok.

Author Response

OK, Thank You.

Reviewer 2 Report

All comments were addressed, but the discussion of elemental doping needs further revision:

the plot in fig. 6c is an EDS spectrum, not an elemental map distribution. A map is what has spatial resolution. Please show an elemental map, or  analyse more EDS spectra from different areas of the same sample and compare the elements concentrations in different areas of the sample. It will not be an evidence of doping, but can be informative to verify the homogeneous In, Nb distribution. 
